# Diagnostic Ability of Methods Depicting Distress of Tumor-Bearing Mice

**DOI:** 10.3390/ani11082155

**Published:** 2021-07-21

**Authors:** Wentao Xie, Marcel Kordt, Rupert Palme, Eberhard Grambow, Brigitte Vollmar, Dietmar Zechner

**Affiliations:** 1Rudolf-Zenker-Institute for Experimental Surgery, University Medical Center Rostock, 18057 Rostock, Germany; marcel.kordt@uni-rostock.de (M.K.); eberhard.grambow@med.uni-rostock.de (E.G.); brigitte.vollmar@med.uni-rostock.de (B.V.); dietmar.zechner@uni-rostock.de (D.Z.); 2Department of Vascular and Thyroid Surgery, Department of General Surgery, The First Affiliated Hospital of Anhui Medical University, Hefei 230022, China; 3Unit of Physiology, Pathophysiology and Experimental Endocrinology, Department of Biomedical Sciences, University of Veterinary Medicine Vienna, 1210 Vienna, Austria; Rupert.Palme@vetmeduni.ac.at; 4Department of General, Visceral, Thoracic, Vascular and Transplantation Surgery, University Medical Center Rostock, 18057 Rostock, Germany

**Keywords:** animal model, animal welfare, xenograft models, in vivo, 3Rs

## Abstract

**Simple Summary:**

Experiments on animals can provide important information for improving the life expectancy and life quality of patients. At the same time, the welfare of these animals is a growing public concern. Therefore, many laws and international guidelines were established with the goal of minimizing the harm inflicted on these animals. A prerequisite of improving animal welfare is to correctly measure how much distress the experiments cause to these animals. However, it is often unknown as to which methods are appropriate to assess distress. Mice bearing subcutaneous tumors are the most frequently used animal model to study the therapeutic effects of drugs. We evaluated if body weight, faecal corticosterone metabolites concentration, burrowing activity and a distress score were capable of differentiating between mice before cancer cell injection and mice bearing large tumors. We observed that only adjusted body weight change and faecal corticosterone metabolites concentration were capable of measuring distress caused by large subcutaneous tumors. Therefore, these two methods are appropriate to assess the welfare of mice with subcutaneous tumors. This knowledge provides a solid basis to optimize animal welfare in future studies. For example, both methods can define the ideal time point when an experiment should end by finding a good compromise between minimal distress for the animals and maximal knowledge gain for mankind.

**Abstract:**

Subcutaneous tumor models in mice are the most commonly used experimental animal models in cancer research. To improve animal welfare and the quality of scientific studies, the distress of experimental animals needs to be minimized. For this purpose, one must assess the diagnostic ability of readout parameters to evaluate distress. In this study, we evaluated different noninvasive readout parameters such as body weight change, adjusted body weight change, faecal corticosterone metabolites concentration, burrowing activity and a distress score by utilising receiver operating characteristic curves. Eighteen immunocompromised NOD.Cg-Prkdcscid Il2rgtm1Wjl/SzJ mice were used for this study; half were subcutaneously injected with A-375 cells (human malignant melanoma cells) that resulted in large tumors. The remaining mice were inoculated with SCL-2 cells (cutaneous squamous cell carcinoma cells), which resulted in small tumors. The adjusted body weight and faecal corticosterone metabolites concentration had a high diagnostic ability in distinguishing between mice before cancer cell injection and mice bearing large tumors. All other readout parameters had a low diagnostic ability. These results suggest that adjusted body weight and faecal corticosterone metabolites are useful to depict the distress of mice bearing large subcutaneous tumors.

## 1. Introduction

According to the World Health Organization, cancer has become the leading cause of death worldwide [1]. Thus, it is necessary to pursue research focusing on this issue. Indeed, many experiments have led to an extraordinary increase in our knowledge of the fundamental molecular mechanisms involving cancer development and improved therapy regimens [2,3]. For example, Trastuzumab, a novel targeted therapeutic regimen, was proven to significantly improve disease-free survival among women with human epidermal growth factor receptor 2 positive breast cancer [4,5]. Furthermore, since 1997, the U.S. Food and Drug Administration has approved many novel drugs for the treatment of several cancers [6]. However, before new treatments for diseases can be developed and introduced into clinical practice, preclinical trials, often using animal experiments, must be conducted [7].

Subcutaneous tumor models using mice are the simplest model for routine evaluation of cancer therapies [2]. The use of human tumor cells, high taking rates, and continuous monitoring are advantages, which make this model an extensively used standard for validation and assessment in oncological studies [8]. In addition, due to an abundance of published data associated with subcutaneous tumor models, it is easy to access the parameters and references needed for testing novel therapies [7].

However, animal experiments are often associated with distress for animals, which has scientific as well as ethical implications [9,10] Distress can affect the results and the conclusions of experiments. For example, it has been proven that distress can affect the outcome of cancer in human patients [11] as well as in animals [12,13,14]. In addition, it is a moral imperative to minimise pain, suffering, distress and harm in experimental animals. These moral issues are addressed by publications, national laws and international guidelines [15,16,17,18,19]. Researchers do have the responsibility and obligation to comply with these laws and guidelines. Unfortunately, the present guidelines and publications only provide little data on distress for specific animal models and on the ability of methods to measure distress. This requires that researchers describe the distress caused by specific cancer models. This might also provide a basis for alleviating the suffering of animals in future experiments. One important first step is to analyse which readout parameters are useful to describe the distress of animals.

Assessing body weight, burrowing activity, faecal corticosterone metabolites (FCMs) and distress scores are commonly used non-invasive methods to quantify distress in mice [20,21,22,23,24]. Body weight loss as an important indicator of distress is used by guidelines as a criterion for euthanasia [2,18,19]. Distress leads to body weight loss and also influences instinctive behaviours in mice, such as burrowing activity [21]. FCMs are a measurement for adrenocortical activity in experimental mice and increased concentration suggests distress [23,24,25]. Moreover, distress scores based on general condition and behaviour can also reflect reduced well-being in mice [17,22,26,27].

The goal of this study was to assess which readout parameters can measure the distress of tumor-bearing mice. In order to assess the diagnostic ability of these readout parameters, we used receiver operating characteristic (ROC) curves.

## 2. Materials and Methods

### 2.1. Cell Culture

The human malignant melanoma cells A-375 (CLS, Eppelheim, Germany) and the cutaneous squamous cell carcinoma cells SCL-2 (donated by Prof. Hahn, University Medical Center Goettingen) were cultured in Dulbecco’s modified Eagle’s medium with high glucose, GlutaMax and pyruvate (Thermo Fisher Scientific, Waltham, MA, USA) supplemented with10% foetal bovine serum (Biochrom GmbH, Berlin, Germany), 100 U/mL penicillin and 100 mg/mL streptomycin (Sigma-Aldrich, Taufkirchen, Germany) in an incubator at 37  °C with 5% CO_2_.

### 2.2. Establishment and Evaluation of the Tumor Model

This study was conducted in accordance with the European directive 2010/63/EU and national law. All experiments were approved by the local ethics committee and the public authority (Landesamt für Landwirtschaft, Lebensmittelsicherheit und Fischerei Mecklenburg-Vorpommern, 7221.3-1-057/18). For this study, 18 male 9–11-week-old NOD.Cg-Prkdcscid Il2rgtm1Wjl/SzJ (NSG) mice were bred in the central animal facility of the Rostock University Medical Center (initially purchased from Jackson Laboratory). The health of the animal stock is routinely checked according to FELASA guidelines (Helicobacter sp., Rodentibacter pneumotropicus, murine Norovirus and rat Theilovirus were detected within the last two years in few animals; these animals were not used for any experiments).

Although we are aware of the debate that the single-housing of mice may cause distress [28,29], we decided that all mice would be single-housed in a Eurostandard Type II L clear plastic cage with a light/dark cycle of 12 h/12 h (dawn: 6:30–7:00 am) and at a temperature of 21 ± 2 °C, with a relative humidity of 60 ± 20%. This was necessary because we needed to assess the burrowing activity and FCM concentrations of single mice. We also provided autoclaved bedding (Bedding Espe Max 3–5 mm granulate, H 0234–500, Abedd, Vienna, Austria), shredded tissue paper (PZN03058052, FSMED Verbandmittel GmbH, Frankenberg, Germany), one paper tunnel (75 × 38 mm, H 0528–151, ssniff, Spezialdiaeten GmbH, Soest, Germany), a wooden enrichment tool (Espe size S, 40 × 16 × 10 mm, H0234.NSG, Abedd, Vienna, Austria), food (pellets, V1534.000, 10 mm, ssniff, Spezialdiaeten GmbH, Soest, Germany) and tap water ad libitum. The bedding material was changed every week.

On day 0, the mice were anaesthetized by 1.5–2.5% isoflurane (CP-pharma, Burgdorf, Germany) in oxygen and placed on a heating plate (37 °C). Then the fur at the flanks of the mice was removed by asid-med (ASID BONZ, Herrenberg, Germany), and 1 × 10^6^ tumor cells mixed on ice with 100 µL 1:1 cold DPBS/Matrigel^®^ High Concentration Growth Factor Reduced (Matrigel^®^ HC GFR, Corning, New York, NY, USA) were injected subcutaneously into the left and right flank, respectively. In order to reduce observer bias, the tumor size was evaluated by scientists during the early (day 4), middle (day 11), and late phase (day 21) in a blinded manner. The longest diameter and the diameter perpendicular to this diameter were measured using a digital caliper (fortis, E/D/E, Wuppertal, Germany) with 0.01 mm precision. Assuming the hemi-ellipsoidal shape of the tumors, the volume was derived using the formula: Volume = 0.52 × Length × Width^2^ [30]. The tumor weight was derived using the formula: Weight = 1.05 × Volume [30]. On day 22, the mice were euthanized by quick anaesthesia with 5 vol % isoflurane, killed with cervical dislocation and the tumors were removed, fixed and embedded in paraffin. Histological sections were then stained with haematoxylin and eosin.

### 2.3. Evaluation of Animal Distress

Before cell injection (data presented as pre-phase), body weight, burrowing activity, faecal corticosterone metabolites (FCMs), and distress scores were assessed on all mice. Throughout the manuscript, we refer to data assessed at this time point as baseline data.

The percentage of body weight change was assessed before cell injection (day 0, pre-phase), after cell injection (day 1, acute phase) and during the early (day 2), middle (day 11) as well as late phase (day 21) of tumor growth (each time point the percentage of body weight change was determined by comparison to the body weight at day –1 before cell injection).

To evaluate the burrowing activity of mice, a tube (length: 15 cm, diameter: 6.5 cm) filled with 200 ± 1 g of food pellets was placed into the cage 3 h before the dark phase (at 04:00–04:10 pm). The mice had free access to these pellets, and the nesting material was left in the cages. The remaining pellets in the burrowing tube were weighed after 17 ± 2 h and the weight of the burrowed pellets was calculated. The burrowing activity was measured before cell injection (day –2, pre-phase), after cell injection (day 0, acute phase) and during the early (day 1), middle (day 10), as well as the late phase (day 20) of tumor growth.

The distress score considers body weight, general condition, spontaneous behaviour, flight behaviour and process-specific criteria as previously published (for details also see Appendix A) [22]. The score was assessed by only one person according to a scoring sheet before cell injection (day –2, pre-phase), after cell injection (day 0, acute phase) and during the early (day 1), middle (day 10) as well as the late phase (day 20) of tumor growth. The distress score was assessed 30 ± 5 min after cell injection on day 0 (on all other days, 09:00–09:30 am).

In order to measure faecal corticosterone metabolites, the bedding with old faeces was removed and replaced by fresh bedding before cell injection (on day −1, pre-phase) and during the late phase of tumor growth (day 20). After 24 h, more than 400 mg fresh faeces were collected. The faeces were dried for 4 h at 65 °C and kept at −20 °C until further processing. Thereafter, 50 mg of dried faeces were extracted with 1 mL 80% methanol and FCMs analysed with a 5α-pregnane-3β,11β,21-triol-20-one enzyme immunoassay [23,31].

### 2.4. Data Presentation and Statistical Analysis

All data were graphed and analysed with GraphPad Prism (version 8.0.1, GraphPad Software Inc., San Diego, CA, USA) and were presented as single data points plus median and 95% confidence interval. The characteristics of data were assessed by the Shapiro–Wilk test. In the case of nonparametric data (distress score, burrowing activity), a one-way repeated measure ANOVA on ranks (Friedman Test) was performed (corrections of multiple comparisons were done as suggested by GraphPad Prism: Dunn’s correction), when analysing the influence of time on the dependent variables (readout parameters). When analysing the influence of the cell lines on the dependent variables, a Mann–Whitney rank sum test (plus Bonferroni correction for multiple time points) was used. In the case of parametric data (tumor size, percentage of body weight change, FCMs, percentage of adjusted body weight change) a two-way repeated measure ANOVA with Geisser–Greenhouse correction (corrections of multiple comparisons were done as suggested by GraphPad Prism: either Sidak test or Tukey test) was performed. Differences with *p* ≤ 0.05 were considered to be significant.

In order to assess the diagnostic ability of each distress parameter, when differentiating between mice with and without tumors, a ROC curve analysis was performed (using data before cell injection and during the late phase of tumor progression). To describe the performance of each readout parameter, the area under the curve, the 95% confidence interval and the P-value were calculated for each parameter. An area under the curve of 1.0 indicates that the parameter is perfect for discriminating between animals growing a tumor and animals not bearing a tumor, whereas a value of 0.5 suggests no discriminative power.

## 3. Results

All tumors were implanted successfully, but the subcutaneous injection of A-375 or SCL-2 cells in mice yielded tumors with completely distinct characteristics. In particular, the volume of A-375 tumors was noticeably larger than the volume of the SCL-2 tumors on day 21 after cell injection (Figure 1a,b). Histological sections of the tumors demonstrated that A-375 cells were densely packed, while in SCL-2 tumors only a few cells were loosely interspersed within the extracellular matrix (Figure 1c,d). After the injection of A-375 cells, the tumor volume significantly increased between the early and middle phase as well as between the middle and late phase of tumor growth (Figure 1e). After injection of the SCL-2 cells, the tumor volume actually decreased between the middle and late phases in all nine mice (Figure 1e). Thus, the volume of A-375 tumors was significantly larger than the volume of the SCL-2 tumors in the middle as well as in the late phase of tumor growth.

In order to evaluate whether mice experience distress after cell injection or during the growth of subcutaneous tumors, body weight change, burrowing activity, a clinical distress score and FCMs were analysed. No loss in body weight or burrowing activity were noticed after cell injection or during tumor growth (Figure 2a,b). Only very low distress scores (maximal distress score of 4 out of 66 theoretically possible points) were observed during the entire experiment (Figure 2c). A moderate increase in the distress score after injection of the A-375 cells and a significant increase in the distress score after the SCL-2 injection was noticed (Figure 2c). Mice had a significantly higher FCM concentration in the faeces in the late phase of A-375 tumor growth when compared to mice before cell injection or compared to mice in the late phase of SCL-2 tumor growth (Figure 2d).

Due to the distinct characteristics of each tumor cell line, the diagnostic ability of each distress parameter was analysed separately. After A-375 cell injection, the diagnostic ability to differentiate between mice with and without tumor was low for body weight change (Figure 3a), burrowing activity (Figure 3b) and distress scores (Figure 3c). However, FCM concentrations (Figure 3d) could differentiate very well between mice with and without tumors. After SCL-2 cell injection, all distress parameters, percentage of body weight change (Figure 4a), burrowing activity (Figure 4b), distress scores (Figure 4c) and FCMs (Figure 4d), could not differentiate between mice with and without a tumor.

Because of the significant volume growth of the A-375 tumors (Figure 1a), adjusted body weight was calculated by subtracting the tumor weight from the body weight (Figure 5). At the late phase of tumor growth, the A-375-injected mice showed a significant adjusted body weight loss (Figure 5a), rather than a body weight gain (Figure 2a). This adjusted body weight had a high diagnostic ability in distinguishing between baseline data and data derived from mice bearing a large A-375 tumor at the late phase (Figure 5b). Adjusted body weight had, however, barely any diagnostic ability in distinguishing between the baseline data and data derived from mice bearing a SCL-2 tumor at the late phase of tumor progression (Figure 5c).

## 4. Discussion

In this study, we could demonstrate that the adjusted body weight change and FCMs have a high diagnostic ability to define distress caused by large subcutaneous tumors. Other readout parameters such as the distress score, burrowing activity or total body weight were not useful to differentiate between mice before cell injection and mice bearing such tumors.

Many studies used total body weight as readout parameter when assessing distress in various animal models [20,26,32]. In our study, this readout parameter was uninformative for differentiating between baseline data and data derived from tumor-bearing mice. This is inconsistent with the widely published concept that severe weight loss can be observed in some animal models during tumor growth [20,33]. However, several other publications also describe that the total body weight was not reduced in tumor-bearing mice [34,35,36,37,38]. For example, no loss of body weight was reported by Husmann et al. when analysing an osteosarcoma mouse model, although metastasis was observed in these mice [38].

However, when we calculated an adjusted body weight, by subtracting the tumor weight from the total body weight, this method could differentiate between mice with and without large tumors. This observation supports the recommendations of some guidelines and is consistent with data presented in some publications. For example, the UKCCCR guidelines emphasize that it is important to focus not only on body weight but also on the weight of the tumor itself [17]. Some authors also mention that the weight of the tumor itself and intraperitoneal effusion may mask the body weight loss [39]. Another study on a mouse xenograft model demonstrated that the size of subcutaneous tumors increased the total body weight of tumor-bearing mice [33]. However, these studies did not compare the adjusted body weight to other readout parameters for distress and did not address the issue if adjusted body weight change has the diagnostic ability to differentiate between mice with and without a tumor.

Please note that we evaluated the body weight changes twice—once as a single parameter and once as part of the distress score. Interestingly, we noticed that the adjusted body weight change has a better diagnostic ability than evaluating a distress score containing body weight change in addition to other parameters (please compare Figure 5b to Figure 3c). This suggests that single readout parameters can have a higher diagnostic ability than complex clinical scores.

FCMs can assess the distress of various animal models and have also been used to evaluate the distress of tumor-bearing mice [35,40,41,42]. In our study, FCM concentrations were not significantly increased when mice had relatively small SCL-2 tumors (<300 mm^3^), but were significantly higher when mice had relatively large A-375 tumors (>1000 mm^3^). In addition, we observed that tumor size moderately correlated with FCM concentration at the late phase of tumor progression (Pearson correlation coefficient of 0.6266, data not shown). We, therefore, conclude that FCMs might only be sensitive enough to indicate distress when mice bear large tumors. 

However, it is a limitation of this study that this conclusion is based on analysing only one mouse strain and two cell lines. One study contradicts our conclusion because it reported that FCM concentrations negatively correlated with tumor size in mice after subcutaneously injecting prostate tumor cells; however, the method to measure FCMS was not validated for use in mice [41]. Our conclusion is supported by several other studies. For example, FCM concentrations were not significantly increased in an orthotopic breast tumor mouse model, when tumors were smaller than 600 mm^3^ [40]. In a mouse model for pancreatic cancer, it was also observed that FCM concentrations did not significantly change when the tumors were small [35]. Thus, our data are consistent with several publications in the literature and support the concept that large subcutaneous tumors can increase FCM concentrations.

Although burrowing activity and clinical distress scores are commonly used as non-invasive methods to assess distress in mice [21,22,27,34], this study failed to define increased distress with these methods during the growth of subcutaneous tumors. Notably, the distress score is often considered to be a subjective readout parameter, which can be influenced by the perception of the researcher. This might be a limitation of this method. However, we also want to mention that similar observations have been published in tumor models for pancreatic cancer and colon cancer [35,43,44]. Possibly, tumor growth in many tumor models causes only mild distress, which cannot be measured by burrowing activity [44] or a distress score [35]. However, it is important to note that some types of cancer can induce, for example, bone pain or severe cachexia [38,45]. Therefore, it cannot be concluded that these methods are uninformative for all cancer models. Indeed, some publications suggest that a clinical distress score or an activity index can be a good indicator to show distress in certain cancer models [34,43].

## 5. Conclusions

Cancer is a general term for a large group of diseases, and the biological characteristics of various types of cancer are quite diverse. Consequently, different cancer models might cause completely different levels of distress. Thus, each cancer model should be carefully evaluated [46]. This study succeeded in defining, that two methods, adjusted body weight and FCMs could define the distress of mice bearing large subcutaneous tumors.. Future studies will have to clarify if these methods are also useful to describe distress caused by cancer in other animal models.

## Figures and Tables

**Figure 1 animals-11-02155-f001:**
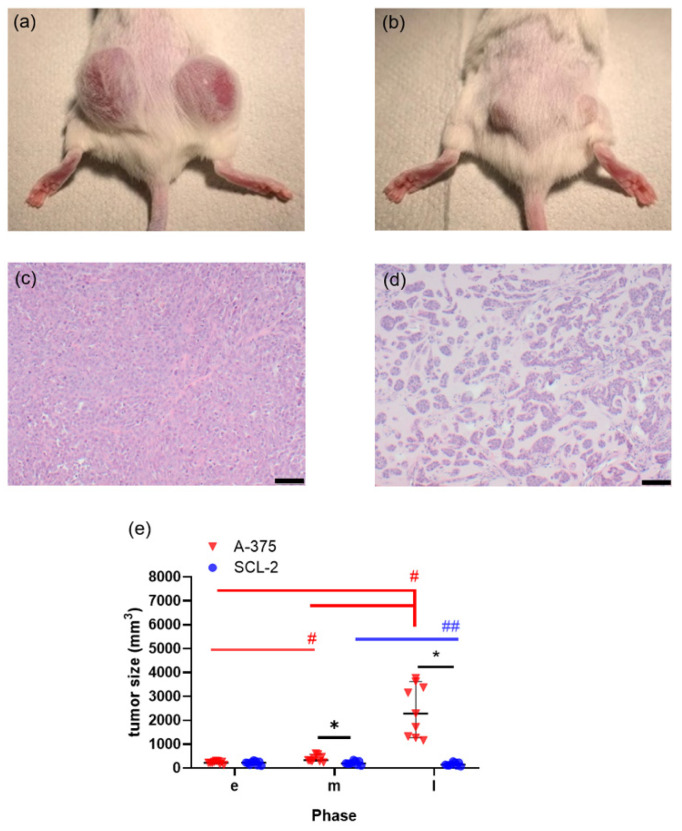
Characterization of A-375 and SCL-2 cell-induced tumors. Morphology of A-375 cell (**a**) and SCL-2 cell (**b**) induced tumors in the late phase after cell injection. Histology of A-375 cell (**c**) and SCL-2 cell tumors (**d**) after hematoxylin and eosin staining; scale bar = 50 μm. (**e**) The sum of tumor size (left flank tumor + right flank tumor) during early (**e**), middle (m) and late (l) phase of tumor growth. * *p* ≤ 0.0116, ^#^
*p* ≤ 0.0091, ^##^
*p =* 0.0003; A-375: n = 9, SCL-2: n = 9.

**Figure 2 animals-11-02155-f002:**
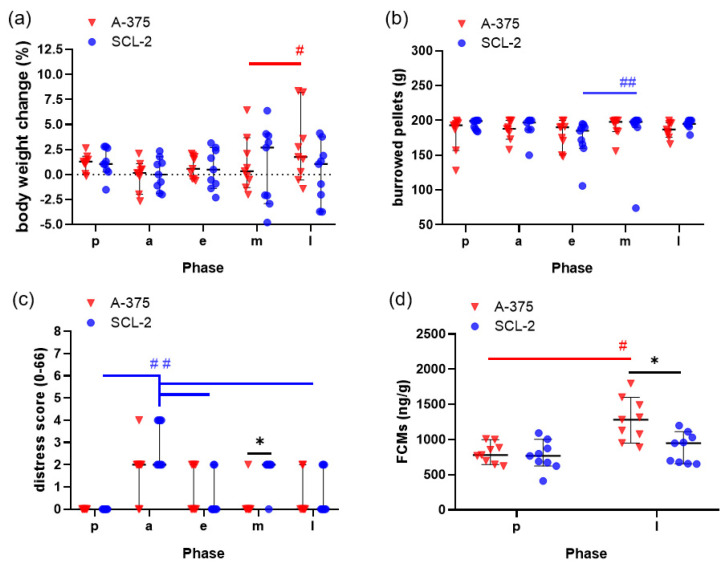
Evaluation of distress during the growth of the A-375 and SCL-2 tumors in mice during pre- (p), acute (a), early (e), middle (m) and late (l) phase. (**a**) Percentage of body weight change, (**b**) burrowing activity, (**c**) distress score and (**d**) concentration of faecal corticosterone metabolites (FCMs). Graph (**a**): ^#^
*p* = 0.0366; (**b**): ^##^
*p* = 0.0287; (**c**): * *p* = 0.0034, ^##^
*p* ≤ 0.0175; (**d**): * *p* = 0.0013, ^#^
*p* < 0.0001; A-375: n = 9, SCL-2: n = 9.

**Figure 3 animals-11-02155-f003:**
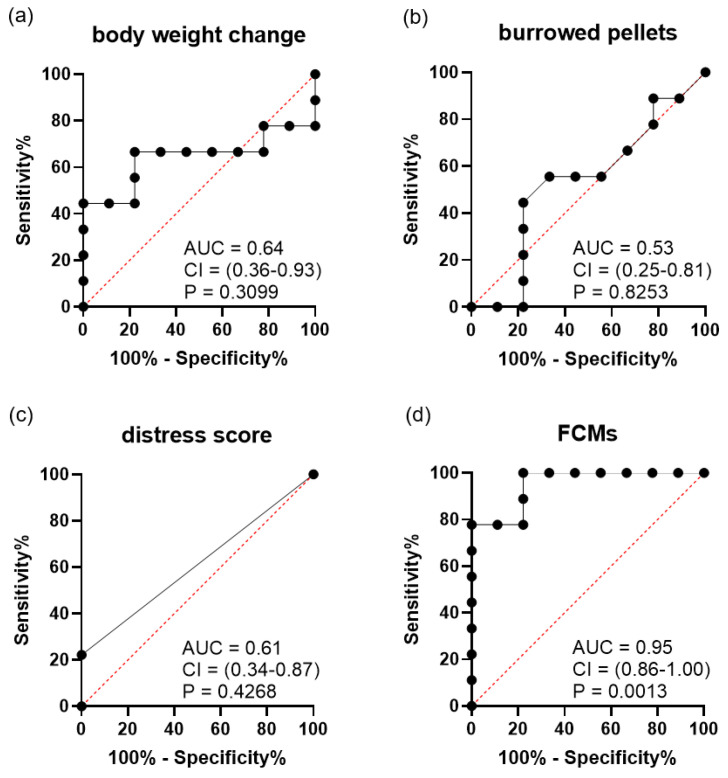
Diagnostic ability of distress parameters when differentiating between mice before cancer cells were injected and identical mice bearing A-375 tumors in the late phase (n = 9). ROC curve analysis of (**a**) percentage of body weight change, (**b**) burrowing activity, (**c**) distress score, (**d**) concentration of faecal corticosterone metabolites (FCMs). The area under the curve (AUC), the 95% confidence interval (CI) and the P-value (P) for testing the AUC to be 0.5 are presented in each graph.

**Figure 4 animals-11-02155-f004:**
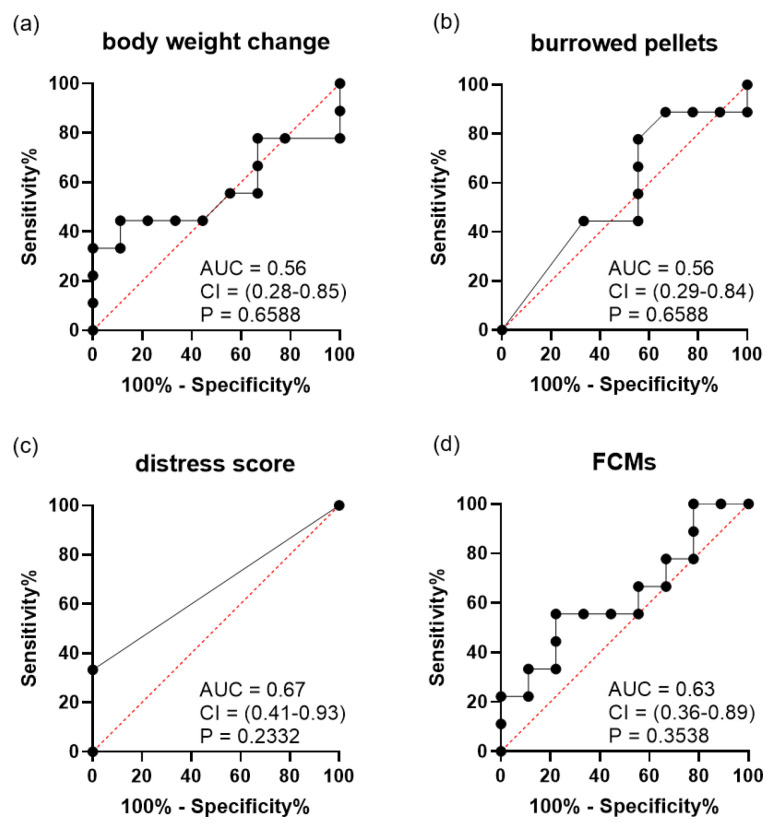
Diagnostic ability of distress parameters when differentiating between the mice before cancer cells were injected and identical mice bearing SCL-2 tumors in the late phase (n = 9). ROC curve analysis of (**a**) percentage of body weight change, (**b**) burrowing activity, (**c**) distress score, (**d**) concentration of faecal corticosterone metabolites (FCMs). The area under the curve (AUC), the 95% confidence interval (CI) and the P-value (P) for testing the AUC to be 0.5 are presented in each graph.

**Figure 5 animals-11-02155-f005:**
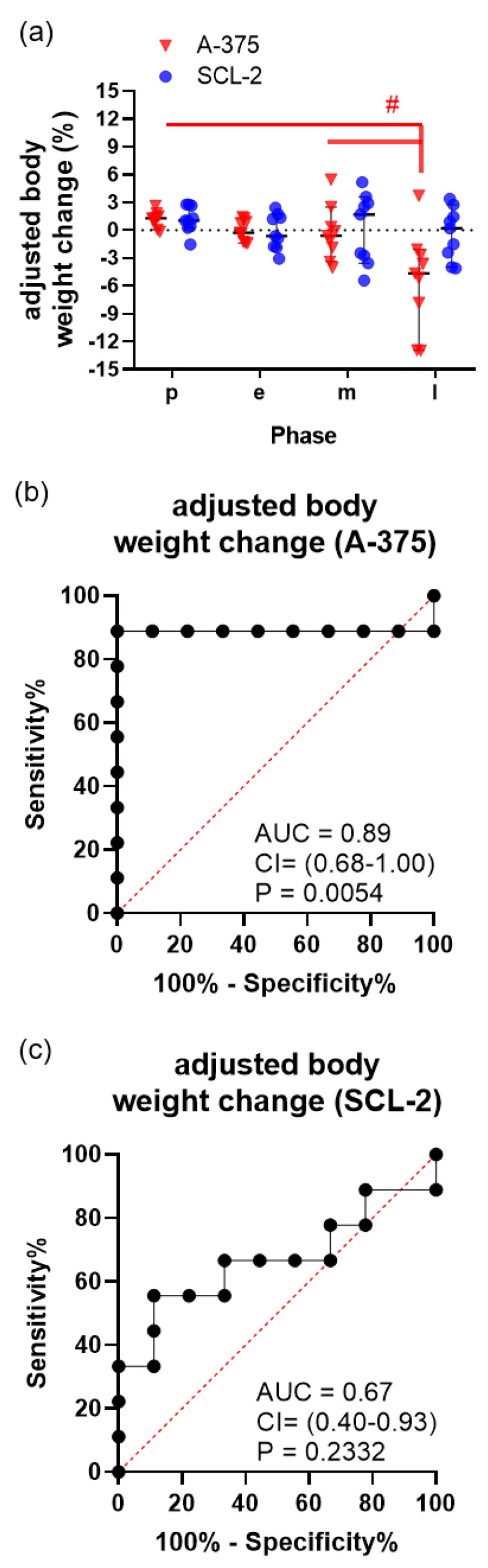
Assessment of adjusted body weight after subtracting the tumor weight from the body weight. (**a**) Percentage of adjusted body weight change of mice during the growth of A-375 and SCL-2 tumors during the pre (p), acute (a), early (e), middle (m) and late (l) phase. Diagnostic ability of adjusted body weight between tumor-free mice and mice with A-375 (**b**) or SCL-2 (**c**) tumors in the late phase. The area under the curve (AUC), the 95% confidence interval (CI) and the *p*-value (P) for testing the AUC to be 0.5 are presented in each graph. ^#^
*p* ≤ 0.0315; A-375: n = 9, SCL-2: n = 9.

## Data Availability

The data presented in this study are available on request from the corresponding author. The data are not publicly available to preserve privacy of the data.

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
