# Peer review of "Diagnostic Ability of Methods Depicting Distress of Tumor-Bearing Mice"

_animals, 2021, doi:10.3390/ani11082155_

Round 1

Reviewer 1 Report

The manuscript by Xie et al. addresses the important point of methodological assessment of distress in mice used in tumor growing research. The study is well described, english writing is clear, figures are fine and the lliterature is sound.

I still have some minor comments and questions:

1.) simple summary: the last sentence addresses an optimization of animal welfare in such models? How could a respective refinement look like?

2.) in the simple summary it sounds like controls were healthy mice. As far as I read it throughout the rest of the manuscript there is a small and large tumor group, no healthy animals, correct?

3.) introduction: minor mistake line 65 - > Researchers

4.) evaluation: line 130: the dark phase is between 07.00p.m. and 07.00a.m.? 12/12 hour dark-light cycle?

Why war burrying evaluted after 17hours +/- 2 hours? Is there a publication of this protocol?

General remark: how stressful would you judge the treatment to induce tumor growth? Could tumor growth affect the circadian rythm of the animals and therefore also affect FCM levels?

Good luck :-)!

Author Response

The manuscript by Xie et al. addresses the important point of methodological assessment of distress in mice used in tumor growing research. The study is well described, english writing is clear, figures are fine and the literature is sound.

I still have some minor comments and questions:

1.) simple summary: the last sentence addresses an optimization of animal welfare in such models? How could a respective refinement look like?

Answer: Future refinement could focuse on using For example, use adjusted body weight instead of total body weight to define humane endpoint in this model.  and FCMs concentration as readout parameter to assess refinement measures. We now describe this concept in the simple summary (line 26-28) using the following text:

This knowledge provides a solid basis to optimize animal welfare in future studies. For example, both methods can define the ideal time point when an experiment should end by finding a good compromise between minimal distress for the animals and maximal knowledge gain for human mankind.

2.) in the simple summary it sounds like controls were healthy mice. As far as I read it throughout the rest of the manuscript there is a small and large tumor group, no healthy animals, correct?

Answer: We apologize to the reviewer that we did not present our experimental design well enough and therefore caused confusion. Indeed, we did not use a separate group of mice without subcutaneous cell injection, but we measured all read out parameters on all mice before (these mice we considered to be healthy) and after cell injection. In order to clarify this point we described (line 130-133) this point in the material and method section.

2.3

Before cell injection (data presented as pre phase), body weight, burrowing activity, faecal corticosterone metabolites (FCMs) and distress scores were assessed on all mice. Throughout the manuscript, we refer to data assessed at this time point as data derived from healthy mice.

3.) introduction: minor mistake line 65 - > Researchers

Answer: we changed Researcher to Researchers on line 68

4.) evaluation: line 130: the dark phase is between 07.00p.m. and 07.00a.m.? 12/12 hour dark-light cycle?

Why was burrowing evaluated after 17hours +/- 2 hours? Is there a publication of this protocol?

Answer: Mice burrows mostly during their active phase (= dark phase), as has been published by Deacon 2012 (Deacon R. Assessing burrowing, nest construction, and hoarding in mice. J Vis Exp. 2012(59):e2607.) We followed the protocol from Deacon 2012, and evaluated the burrowing activity during this entire dark phase, as we have done so in our previous publications (see reference Tang G, Seume N, Häger C, et al. Comparing distress of mouse models for liver damage. Sci Rep. 2020;10(1):19814.)

General remark: how stressful would you judge the treatment to induce tumor growth? Could tumor growth affect the circadian rythm of the animals and therefore also affect FCM levels?

Answer: Actually, Subcutaneous tumor cells injection have been suggested to cause mild distress (see reference Directive 2010/63/EU of the European Parliament and of the Council of 22 September 2010 on the protection of animals used for scientific purposes. Section III, 1(g)). The data of our study agree with this suggestion, because no or very little distress was observed during the acute phase (Figure 2). We agree with reviewer that large tumors could affect the circadian rhythm and might, therefore, indirectly influence FCM concentrations. However, we have no way of differentiation between a direct effect of large tumors on distress and an effect via disruption of the circadian rhythm. In addition, we argue that also the disruption of the circadian rhythm might be interpreted as distress for the animals.

Reviewer 2 Report

Diagnostic ability of methods depicting distress of 2 tumor-bearing mice

Xie et al describe an investigation to determine the useful of various parameters related to distress in mice to identify those with the greatest potential to detect early signs of declining welfare (distress) during tumour development. To accomplish this, they tested two different types of heterotopically implanted tumour cell lines in two groups of nine NSG (immunocompromised) mice. The parameters measured were body weight (adjusted and not), a distress score, burrowing activity and an assessment of faecal corticosterone levels.

They found that compared to the SCL-2 (squamous cell carcinoma) cell line, mice bearing A-375 (human melanoma) tumours were significantly more likely to suffer stress when tumours became large. They found that body weight adjusted for tumour burden and faecal corticosterone were the parameters with the greatest diagnostic ability for determining the presence of distress.

Apart from some minor typographical and grammatical errors the manuscript is reasonably well written, and the findings are well presented. However, the report lacks detail and potentially suffers from a major study design issue.

As far as the lack of detail is concerned the major issue is that they have not adequately described how the distress scoring procedure was conducted, and what elements it contained. It is said that this had a maximum score of 66, but how that maximum score might be arrived at is never mentioned. Because of this, it is impossible to determine whether their conclusion that was not useful for distress detection becomes impossible. Section 2.3 says that apart flight behaviour (which is?), and process-specific criteria (which are not defined or described), the distress score considered body weight. That being the case, why was body-weigh assessed as separate from the distress score? Apart from not knowing what this scoring entailed, it is not said who conducted it. If this was done by more than 1 person, likely being a subjectively orientated method, then it is highly likely that this could have been biased. Also, if more than one person did this scoring then the study requires an analysis of reliability and validity. Furthermore, without this information it is not possible to evaluate the strength of the ROC analysis (which is also not adequately explained). Please include an example scoresheet and include instructions on its application.

As to the study design, there is no mention of any justification of animal numbers, which appear rather low at n=9. However, what is most concerning is the lack of an appropriate control group. To state that mice bearing large tumours are substantially more prone to distress one needs to compare the findings with a healthy control group. On that point, they frequently refer to comparison with healthy mice when in fact they do not have any. While reading the manuscript I presumed this meant mice in which tumours failed to develop, because all 18 were originally implanted with some type of tumour material. How many mice that was is not mentioned and is an essential piece of information because it effectively lowers the number of tumour bearing mice with which to assess the value of the different distress scoring parameters (lowering the study power even further). The manuscript requires a report of how many mice were or were not successfully engrafted, and to assign those failures as healthy is wrong. As it would be a shame that these mice have been wasted, in this instance, and if they did not originally do so, a retrospective power evaluation would be marginally acceptable. As it stands, it is impossible to agree with the major conclusions. When you do not know the impact of either cell line relative to normality then you cannot conclude that impact was minor.

Other points:

The mice were housed singly, which is not only a potential confound but one that could also have impacted on stress susceptibility. Presumably, this was done to enable the assessment of burrowing behaviour and collection of faecal samples? If undertaking that, why not also score next complexity to further justify single housing? The most important outcome was that faecal corticosterone predicted predictor of distress in mice bearing large tumours. Why was this only done twice, at the beginning and end of the study. No doubt the reply will be that they did not want to disturb the mice as the study progressed. However, there should have been at least one cage clean between the beginning and end of the study, providing an opportunity to collect at least of extra set of samples. If not, did you really leave the cages dirty for 20 days? Also, once the old faeces were removed, how long were the mice given to defecate before the relevant samples were collected. 0.4 grams is a lot for one mouse. In any case, although the nature of it is unclear, the mice must have been disturbed to some extent during the distress scoring process, so it would be important to know how this impacted the mice; providing another layer of assessment towards evaluating the ‘actual severity’ of the procedures combined. (see here for assistance with this: https://ec.europa.eu/environment/chemicals/lab_animals/pdf/examples.pdf).

Whatever the reason for single housing the justification for this has to be explained.

Minor issues:

Simple Summary: Remove the comma after the word ‘measure’. On the next line change ‘animal’ to animals.

Abstract: Provide mouse strain details and define tumour types.

Introduction: Change ‘regiment’ to regimen.

Section 2.2: Presumably, you meant these pathogens were ‘not’ detected rather than detected?

Section 2.3: As above, what are process-specific criteria and flight behaviour?

Section 2.4: Why re these values assessed non-parametrically. The evaluation of burrowing materials, in grams, is a scale variable? Instead of receiver operating characteristic, define earlier and refer to this as ROC. State here how many mice failed to engraft and why you deemed these to be ‘healthy’.

Results: How do explain that in some instances the SCL-2 tumours shrank, and in what proportion of the group? As above, what are the 66 possible points on the distress score?

Figure 3: The legend mentions the differentiation between tumour free mice and those with large tumours. As above, how many?

Discussion: You say that the weight data were ‘consistent with several publications describing that total body weight was not reduced in tumor bearing mice’. This is not true, unless you say you are referring to mice with subcutaneous tumours only. Mice with internalised cancer often lose weight.

Round 2

Reviewer 2 Report

The author revisions have adequately addressed the majority of the issues. The inclusion of the distress scoring sheet is very helpful. However, the revised manuscript still does not address the confusion I had about the statements about comparisons with 'healthy' mice. The revised figure legends now state '....when differentiating between tumor-free mice (n=9) and mice with A-375 tumors (n=9) in the late phase.' This makes it even more likely that reader will think there are 18 mice split between two groups.

You need to make it abundantly clear that the 'healthy' data were those collected at the beginning, and I suggest to refer to these as baseline data that were considered to represent those from healthy mice. 

In the last line of the abstract delete 'for human humankind'.  

Author Response

The author revisions have adequately addressed the majority of the issues. The inclusion of the distress scoring sheet is very helpful. However, the revised manuscript still does not address the confusion I had about the statements about comparisons with 'healthy' mice. The revised figure legends now state '....when differentiating between tumor-free mice (n=9) and mice with A-375 tumors (n=9) in the late phase.' This makes it even more likely that reader will think there are 18 mice split between two groups.

Point 1: You need to make it abundantly clear that the 'healthy' data were those collected at the beginning, and I suggest to refer to these as baseline data that were considered to represent those from healthy mice. 

Reply to point 1: To eliminate confusion, all ‘healthy mice’ were replaced in whole manuscript. (see line 21, line 40, line 241-244, line 258, line 262, line 278)

We evaluated if body weight, faecal corticosterone metabolites concentration, burrowing activity and a distress score were capable of differentiating between mice before cancer cell injection and mice bearing subcutaneous tumors.

Adjusted body weight and faecal corticosterone metabolites concentration had a high diagnostic ability in distinguishing between mice before cancer cell injection and mice bearing large tumors.

Throughout the manuscript, we refer to data assessed at this time point as baseline data.

This adjusted body weight had a high diagnostic ability in distinguishing between baseline data and data derived from mice bearing a large A-375 tumor at the late phase. Adjusted body weight had, however, barely any diagnostic ability in distinguishing between baseline data and data derived from mice bearing a SCL-2 tumor at the late phase of tumor progression.

Other readout parameters such as the distress score, burrowing activity or total body weight were not useful to differentiate between mice before cell injection and mice bearing such tumors.

In our study this readout parameter was uninformative for differentiating between baseline data and data derived from tumor bearing mice.

However, these studies did not compare the adjusted body weight to other readout parameters for distress and did not address the issue if adjusted body weight change has the diagnostic ability to differentiate between mice with and without a tumor.

To describe some figures more clearly, we rewrote the figure legend of figure 3 and 4.

Diagnostic ability of distress parameters, when differentiating between mice before cancer cells were injected and identical mice bearing A-375 tumors in the late phase (n=9).

Diagnostic ability of distress parameters, when differentiating between mice before cancer cells were injected and identical mice bearing SCL-2 tumors in the late phase (n=9).

Point 2: In the last line of the abstract delete 'for human humankind'.

Reply to point 2: Sorry for the spelling error. We changed ‘human humankind’ to ‘mankind’. (see line 28)